# Pharmacomicrobiomics of Classical Immunosuppressant Drugs: A Systematic Review

**DOI:** 10.3390/biomedicines11092562

**Published:** 2023-09-18

**Authors:** Annalaura Manes, Tiziana Di Renzo, Loreta Dodani, Anna Reale, Claudia Gautiero, Mariastella Di Lauro, Gilda Nasti, Federica Manco, Espedita Muscariello, Bruna Guida, Giovanni Tarantino, Mauro Cataldi

**Affiliations:** 1Section of Pharmacology, Department of Neuroscience, Reproductive Sciences and Dentistry, Federico II University of Naples, 80131 Naples, Italy; annalaura.manes@gmail.com (A.M.); loretadodani@hotmail.com (L.D.); fede.manco3@hotmail.com (F.M.); 2Institute of Food Sciences, National Research Council, 83100 Avellino, Italy; tiziana.direnzo@isa.cnr.it (T.D.R.); anna.reale@isa.cnr.it (A.R.); 3Physiology Nutrition Unit, Department of Clinical Medicine and Surgery, Federico II University of Naples, 80131 Naples, Italy; claudia.gautiero@gmail.com (C.G.); msdl@hotmail.it (M.D.L.); gildanasti@libero.it (G.N.); bguida@unina.it (B.G.); 4Nutrition Unit, Department of Prevention, Local Health Authority Napoli 3 Sud, 80059 Naples, Italy; edy.muscariello@gmail.com; 5Department of Clinical Medicine and Surgery, Federico II University of Naples, 80131 Naples, Italy; giovanni.tarantino@unina.it

**Keywords:** corticosteroids, cyclosporine, everolimus, microbiota, methylprednisolone, mycophenolic acid, prednisolone, prednisone, sirolimus, tacrolimus

## Abstract

The clinical response to classical immunosuppressant drugs (cIMDs) is highly variable among individuals. We performed a systematic review of published evidence supporting the hypothesis that gut microorganisms may contribute to this variability by affecting cIMD pharmacokinetics, efficacy or tolerability. The evidence that these drugs affect the composition of intestinal microbiota was also reviewed. The PubMed and Scopus databases were searched using specific keywords without limits of species (human or animal) or time from publication. One thousand and fifty five published papers were retrieved in the initial database search. After screening, 50 papers were selected to be reviewed. Potential effects on cIMD pharmacokinetics, efficacy or tolerability were observed in 17/20 papers evaluating this issue, in particular with tacrolimus, cyclosporine, mycophenolic acid and corticosteroids, whereas evidence was missing for everolimus and sirolimus. Only one of the papers investigating the effect of cIMDs on the gut microbiota reported negative results while all the others showed significant changes in the relative abundance of specific intestinal bacteria. However, no unique pattern of microbiota modification was observed across the different studies. In conclusion, the available evidence supports the hypothesis that intestinal microbiota could contribute to the variability in the response to some cIMDs, whereas data are still missing for others.

## 1. Introduction

Classical immunosuppressant drugs (cIMD), which include glucocorticoids (GCs), cyclosporine (CyA), tacrolimus (TAC), the mammalian target of rapamycin (mTOR) inhibitors sirolimus (SIR) and everolimus (EVERO), and the Inosine Monophosphate Dehydrogenase (IMPDH) inhibitor mycophenolic acid (MPA) and its prodrug mycophenolate mofetil (MMF), still have a central role in transplant medicine since they represent the cornerstone of the pharmacological treatment to prevent and treat organ rejection [1]. In addition, these drugs are also used for the treatment of autoimmune diseases in selected circumstances in which biological immunosuppressant drugs are not indicated. The use of cIMDs in clinical practice is complicated by their narrow therapeutic window, which makes it difficult to achieve good immunosuppression without toxicity [2]. A further element complicating the use of many cIMDs is their unpredictable pharmacokinetics with large variations in the plasma concentrations that are attained when similar doses are given to different patients [3,4,5]. This problem has traditionally been addressed by performing therapeutic drug monitoring (TDM), assessing the concentration of cIMDs in the blood and then adjusting their doses according to the results obtained [6]. Whilst this approach improves safety and efficacy of the treatment, it is far from optimal due to the empirical “fail and correct” approach, which implies that drug plasma concentrations could be dangerously higher (or lower) than optimal for a while before being corrected. As a matter of fact, organ rejection and cIMD toxicity also occur in patients undergoing accurate TDM [7]. It would be, therefore, highly desirable to identify difficult patients early to design their therapeutic plan on a personalized basis. Even though it has not yet been routinely implemented yet in transplant medicine, pharmacogenotyping might help achieve this goal by identifying patients with polymorphisms in genes encoding key enzymes involved in immunosuppressant drug metabolism [8]. A new factor, only recently identified, that could be responsible for the interindividual variability in drug response is the patient’s microbiota, i.e., the whole population of microorganisms living in a commensal status in humans (or animals) [9]. These microorganisms are metabolically active and able to affect host health in different ways, such as processing of nutrient molecules, modulating immune response in the gut and liver, and the generating small molecule mediators acting systemically, after being absorbed into the general circulation [10]. Importantly, human microbiota may also affect patient response to pharmacological therapy [11]. The term *Pharmacomicrobiomics* has been coined, in consonance with the already existing term *Pharmacogenomics*, to design the whole array of interactions by which microbiota may modify host response to pharmacologically active substances [12,13]. Some evidence that bacteria of the intestinal microflora could metabolize specific drugs and, therefore, affect their efficacy was already obtained long time ago, and probably the first documented case of such an interaction is the conversion of the inactive dye prontosil red into the potent antibacterial compound sulphanilamide, reported in 1937 [14]. Thereafter, several other drugs such as digoxin have been demonstrated to be victims of bacterial metabolism in the gut. In addition, pharmacological strategies have been developed to exploit the metabolizing capacities of intestinal bacteria in the design of new drugs. More specifically, the ability of bacteria selectively located in the colon to perform azoreduction reactions has been used to design drugs intended to be specifically active in the colon such as sulphasalazine and its congeners, which still have an important pharmacological role in the treatment of ulcerative colitis [12]. Whilst what has just been mentioned represents well acquainted issues in the pharmacology of selected medicines, much more recently the interest in microbial metabolism of drugs has reemerged strongly since it has been realized that it may contribute to the interindividual variability in the response to a wider group of drugs than originally suspected [15] either by exerting a pre-systemic metabolism or by accumulating some of these drugs (e.g., duloxetine, montelukast, rosiglitazone and roflumilast) in their cytoplasm and, therefore, preventing their absorption and reducing their bioavailability and, possibly, clinical efficacy [16]. Therefore, it has been suggested that microbiota composition should be included among the factors to be evaluated in precision medicine [17] also considering that the composition of human microbiota varies interindividually in healthy subjects, and it may be severely modified in specific diseases [18]. These considerations suggest that part of the interindividual variability in the clinical response to cIMDS could be dependent on patient microbiota. Therefore, in the present paper we performed a systematic review of the published studies on the impact of intestinal microbiota on orally given cIMDs with the aim of evaluating the strength of evidence in support of this hypothesis.

## 2. Materials and Methods

The protocol of this systematic review has been registered in the International Platform of Registered Systematic Review and Meta-analysis Protocols (INPLASY, Middletown, DE, USA) with registration number is INPLASY202380129 and is freely available at the address: https://inplasy.com/?s=INPLASY202380129 (accessed on 31 August 2023).

### 2.1. Purpose

The present systematic review addressed the following two research questions:Does the intestinal microbiota affect cIMD pharmacology (pharmacokinetics, efficacy or tolerability)?Do cIMDs modify the composition of intestinal microbiota?

The following Section 2.2, Section 2.3, Section 2.4 and Section 2.5 report the PICO (Population–Intervention–Comparison–Outcome) framework of our systematic review.

### 2.2. Population

Preclinical studies performed in healthy animals or in animals with experimental organ transplantation or with autoimmune diseases receiving cIMDs, and clinical studies conducted in humans treated with these drugs because recipients of organ transplantation or because affected with autoimmune diseases were considered for the review. Studies with one or more of the following features were evaluated for being included in the revision:Metagenomic characterization of fecal or ileal microbiota,Pharmacokinetics analysis of the administered immunosuppressant drugs,Analysis of cIMD efficacy and/or toxicity, andAnalysis of the effect of wide-spectrum antibiotics on points from 1 to 3.

We also considered studies using reconstituted systems in vitro that examined the effect of fecal extracts/bacterial cultures/bacterial enzymes on cIMDs.

### 2.3. Intervention

The present systematic review only included studies in which the following cIMDs were given orally in various combinations: CyA, EVERO, GCs, MMF, MPA, SIR or TAC.

### 2.4. Comparison

We considered for inclusion in the review both studies in which a control group (vehicle treated animals in preclinical experimentations or age-matched healthy controls in clinical studies) was compared with a group receiving cIMDs, and pre-, post-studies in which the same animals/individuals were compared before and after the treatment with these drugs.

### 2.5. Outcome

The outcomes of the studies considered in the present review were changes in the efficacy/tolerability of cIMDs and/or in their pharmacokinetic parameters.

### 2.6. Information Sources and Search Strategy

A systematic literature search was performed through 3 December 2022 using the PubMed (https://pubmed.ncbi.nlm.nih.gov/) and Scopus (https://www.scopus.com/) databases and the following keywords in various combinations: cyclosporine, tacrolimus, sirolimus, everolimus, mycophenolic acid, mycophenolate mofetil, prednisone, methylprednisone, pharmacomicrobiomics, gastrointestinal microbiome, metabolism, microbes, and bioaccumulation. No specific limit was posed on the year of publication. Only original English language studies were considered for review; narrative reviews, systematic reviews, metanalyses, guidelines, consensus papers, case reports, editorials and commentaries were all excluded.

### 2.7. Data Extraction

The database search was performed independently by A.M. and L.D. A first screening of the papers retrieved was performed by A.M. and M.C. after reading and examining their titles and abstracts. Selected papers underwent further selection by A.M. and M.C. upon reading and examining the full text of the articles.

### 2.8. Risk of Bias in Individual Studies

The risk of bias of animal studies was evaluated using the SYRCLE’s risk of bias tool for animal studies, which is and adaption to animal studies of the RoB2 tool [19].

The risk of bias of observational studies in humans was evaluated using the Grade Criteria which consider five different categories of bias: (1) Inappropriate eligibility criteria; (2) Inappropriate methods for exposure and outcome variables; (3) Not controlled for confounding; (4) Incomplete or inadequate follow-up; (5) Other limitations [20]. For all the observational human studies the risk related to the aforementioned risk categories was conjunctly rated by A.M. and M.C. as either “no risk”, “unclear risk” or “high risk” and the results obtained were reported in a tabular format both as average percentage for the whole group of papers and as individual data for the single studies.

The risk of bias of randomized trials was assessed using the five categories of the Grade Criteria (lack of allocation concealment, lack of blinding, incomplete accounting of patients and outcome events, selective outcome reporting, and other limitations). As for observational studies, the risk for each of these categories was conjunctly rated by A.M. and M.C. as either “no risk”, “unclear risk” or “high risk”.

### 2.9. Data Synthesis

The main findings of the selected studies were summarized in a narrative form in the text of the article and in two tables, the first (Table 1) concerning the papers on the effect of the intestinal microbiota on cIMDs [21,22,23,24,25,26,27,28,29,30,31,32,33,34,35,36,37,38,39,40], and the second (Table 2) concerning the papers on effect of cIMDs on intestinal microbiota [41,42,43,44,45,46,47,48,49,50,51,52,53,54,55,56,57,58,59,60,61,62,63,64,65,66,67,68,69,70]. The following variables were extracted and included in Table 1: 1—lead author and year of publication, 2—the experimental model used, 3—the drugs tested, 4—the presence/absence of changes in the pharmacokinetic properties of the tested drugs, and 5—a detailed description of the effect of intestinal microbiota on cIMD pharmacokinetics. The variables extracted to prepare Table 2 were: 1—lead author and year of publication, 2—the species (human, mice, rats) object of the experimentation, 3—the condition/disease either spontaneous or experimental that was treated with cIMDs, 4—the specific cIMDs that were used with doses, administration modalities and duration of treatment, 5—the presence or absence of intestinal microbiota modifications upon treatment with cIMDs, and 6—a detailed description of the changes occurring in intestinal microbiota including, when available, data on α- and β-diversity and taxonomic information at the level of phylum, class, order, family, genus, and species. No metanalysis of the data was considered feasible considering the heterogeneity in the research questions addressed, in the species and in the experimental models used in the experimentations and in the drugs administered.

## 3. Results

### 3.1. Search Results

The results of the literature search that we performed are summarized in Figure 1 which shows the PRISMA flowchart of this study. A total of 1055 published papers were retrieved at the initial database search: 225 were obtained by searching PubMed and 830 searching Scopus. Four hundred thirty-seven records were duplicated in the two databases and, therefore, after correcting for duplication, the total number of papers that were considered for title and abstract analysis was 618. After excluding the publications not complying with inclusion criteria, 86 records were considered suitable for full-text analysis. After reading these papers and examining their content the two reviewers identified 50 papers [21,22,23,24,25,26,27,28,29,30,31,32,33,34,35,36,37,38,39,40,41,42,43,44,45,46,47,48,49,50,51,52,53,54,55,56,57,58,59,60,61,62,63,64,65,66,67,68,69,70] as suitable for inclusion in the systematic review, whereas the remaining 36 [71,72,73,74,75,76,77,78,79,80,81,82,83,84,85,86,87,88,89,90,91,92,93,94,95,96,97,98,99,100,101,102,103,104,105,106] were excluded since they were not compliant with one or more of the inclusion or exclusion criteria.

Seven of the reviewed studies were entirely performed in vitro [23,25,26,30,31,36,37], twenty-three reported the results of studies performed in vivo only in animal models, eight in rats [39,42,44,50,51,61,62,70] and fifteen in mice [43,45,46,47,48,49,52,53,57,58,63,66,67,68,69], whereas one paper included the results of experiments performed in both mice and humans [34]. Nineteen papers were investigations performed only in humans. Eighteen of them were observational studies [21,22,24,27,28,29,32,33,35,38,40,41,54,55,56,60,64,65] and only one was a randomized clinical trial [59].

### 3.2. Quality Assessment

None of the reviewed animal studies fully complied with the SYRCLE’s requirement for risk of bias assessment [19]. In particular, no information was reported on the methods of randomization and of allocation concealment (selection bias), random housing and blinding of caregivers/investigators (performance bias), random selection of the animals for assessment and blinding of the assessors (detection bias), and strategies used to handle incomplete data (attrition bias). Therefore, for all these categories, we assigned the score “uncertain risk of bias”. Conversely, no apparent risk of bias was found for group similarity of at baseline (selection bias) and for selective outcome reporting (reporting bias).

The risk of bias of the eighteen observational studies in humans was assessed according to the Grade Criteria for observational studies (see methods for more details). Figure 2 reports a summary of the results of our quality assessment shown as the percentage of studies rated as “low”, “unclear” and “high” risk of bias for each of the five categories of bias considered in the Grade Criteria.

We identified a high risk of bias in more than 60% of the studies for two of these categories: *Inappropriate eligibility criteria* and *Not controlled for confounding*. In addition, we rated the risk of bias for eligibility criteria as “unclear” in approximately 30% of the studies (mostly retrospective investigations) since no clear inclusion and/or exclusion criteria were reported besides identifying the disease under evaluation. Only 11% of the studies included a correction for confounding factors and, therefore, were rated as “no risk of bias”; in approximately 28% of the studies, the risk originated by the lack of correction was unclear since no clear difference in covariates emerged from reported data, whereas in the remaining 61% of the studies, we rated the bias risk as high because of major disparities among groups in variables such as gender or pharmacological treatment with drugs other than cIMDs that were not corrected for. Seven studies had a cross-sectional design and, therefore, did not include any follow-up of the recruited patients. In six of the remaining eleven prospective studies the follow-up was complete with no risk of bias, whereas in five we identified some incompleteness in the follow-up with unclear consequences on the risk of bias. We identified critical points for the category *Methods for exposure and outcome*, whose implications for bias risk were major in two studies and unclear in six.

Figure 3 reports the detailed score for the various categories of risk of bias in each of the eighteen reviewed human observational studies examined.

The study by Pigneur et al. [59] was the only randomized clinical trial among the papers included in the present systematic review. At the bias risk assessment, we did not find information about allocation concealment, which, therefore, we scored as “unclear”. Blinding was reported for the assessors, whereas it was not declared but probably impossible (due to the type of intervention, exclusive enteral nutrition versus GCs) for patients. We did not find any evidence of incomplete accounting of patients and outcome events, of selective outcome reporting, or of other limitations and, therefore, we scored these three categories as free of risk.

### 3.3. Effect of Intestinal Microbiota on cIMDs

We reviewed 20 papers addressing our research question 1: Does the intestinal microbiota affect cIMD pharmacology (pharmacokinetics, efficacy or tolerability)? Table 1 summarizes the main findings of these studies. Eight studies were on MMF, ten on TAC, four on CyA, and ten on GCs. We found no studies on the effect of the intestinal microbiota on EVERO or SIR. Seventeen studies showed significant effect of fecal microbiota, or enzymes produced by intestinal bacteria on cIMDs [21,22,23,25,26,27,28,29,30,31,32,33,34,35,38,39,40], whereas no effect was observed in the remaining three studies [24,36,37].

Seven papers reported results of studies performed in vitro either by using living cells (cell lines or bacteria) or acellular systems with cell extracts or purified enzymes [23,25,26,30,31,36,37], twelve papers were on studies performed in vivo [21,22,24,27,28,29,32,34,35,38,39,40] and one included both in vitro ed in vivo experiments [33]. Seven of the in vitro studies evaluated cIMD metabolism either by cultured fecal bacteria from human stool (two papers [25,26]), or by human fecal material (three papers [33,36,37]) or by purified enzymes (two papers [30,31]). One paper investigated the effect of deoxycholic acid (DCA) and chenodeoxycholic acid (CDCA), generated from bile salts by bacterial metabolism, on the activity of ABC-B1, a pump responsible for CyA efflux from intestinal epithelial cells [23]. The main findings of the in vitro studies were that: (1) GCs can be degraded by fecal bacteria through the action of desmolases (DesA and DesB) [26,30,31], (2) TAC is converted to the less active metabolites M1/M2 by the bacterial enzymes TacA and TacB [25]; (3) The main metabolite of MMF, mycophenolate glucuronide (MPAG) is deglucuronated by bacterial β-Glucuronidase (GUS) to MPA [26]; (4) MFA is not degraded by fecal microbiota [26]; (5) CyA is not significantly metabolized by intestinal bacteria [26,37] but DCA and CDCA, generated from bile salts by bacterial metabolism, increase its cytotoxicity impairing ABC-B1 activity [23]; (6) data are missing for EVERO and SIR.

Of the thirteen in vivo studies, eleven were performed in human patients who underwent kidney [24,28,29,33,38], heart [21,27,34] or hemopoietic stem cell transplantation [22,32] or were affected with ulcerative colitis (UC) [40], one study in rats and one in both mice and in humans [34]. Five of the ten studies performed in humans investigated the effect of the intestinal microbiota on cIMD toxicity (four on diarrhea [24,29,33,38] and one on neutropenia [21]). Of the remaining six studies, two were on cIMD pharmacokinetics (one on MPA, performed in humans [32] and the other on CyA, performed in rats [39]), and four on cIMD efficacy (two on the need for TAC dose escalation [27,28], one on GC responsiveness [40], and two on the occurrence of graft vs. host disease (GvHD), as a consequence of immunosuppressive therapy failure [22,35]). The main findings of these studies were: (1) The intestinal microbiota is different in patients developing or not neutropenia during the immunosuppressant therapy with MMF and TAC [21]; (2) the composition of the intestinal microbiota is different in patients who develop or not acute GvHD after allogenic hemopoietic stem cell transplantation [22] and in patients with acute GvHD responding or not to PRED; (3) the conversion of MPAG back to MPA, and, consequently, MPA enterohepatic recirculation, depend on the composition of the intestinal microbiota [32]; (4) the composition of the intestinal microbiota is different in patients who develop or not diarrhea during treatment with MMF-containing regimens [29,38] and bacterial GUS activity correlates with the risk of developing MMF-induced diarrhea [33,34,38]; (5) the composition of the intestinal microbiota is different in patients requiring low or high TAC doses [27,28]; (6) the composition of the intestinal microbiota affects CyA bioavailability in rats [39]. The previous points are just a list of the main findings of the studies detailed in Table 1. In fact, we could not produce a real synthesis of the data since, for each of the cIMDs, we retrieved only very few papers, that, moreover, used, heterogeneous experimental techniques therefore hindering any joint analysis.

### 3.4. Effect of cIMDs on Intestinal Microbiota

We reviewed 32 papers addressing our research question 2: Do cIMDs modify the composition of intestinal microbiota? Table 2 summarizes the main findings of these studies. Twenty-three papers reported the results of studies performed in experimental animals, seven in rats [42,44,50,51,61,62,70] and sixteen in mice [34,43,45,46,47,48,49,52,53,57,58,63,66,67,68,69]. The remaining nine studies were conducted in humans who received cIMDs either because they were recipients of organ transplantation (LT in two studies [41,64], KT in two [33,65] and HSCT in one [56], or because affected with autoimmune diseases (autoimmune pancreatitis [54], nephrotic syndrome [55], Crohn disease [59], or transverse myelitis [60]). In most of the studies performed in animals only one cIMD was used, even though in some of them different groups of animals each receiving a different cIMD were studied. In six of these studies TAC was investigated [42,46,51,52,66,69], in seven SIR [42,43,47,49,53,58,68], in six MMF [34,45,57,61,62,66], in six GC (in four PRED [48,66,67,70] and in two PREDN [44,63]), in one CyA [50] and in another one EVERO [66]. One study performed in normal mice evaluated not only the effect of PRED, TAC, EVERO and MMF given alone but also of the combined treatment with PRED, TAC, and MMF [66]. In four of the nine studies performed in humans only a cIMD was used, and more specifically it was a GC (PRED in three papers [55,59,60] and PREDN in one [54]). In the remaining five papers, patients were treated with multiple cIMDs as part of multidrug combination immunosuppressive regimens: TAC was given in four of these five studies [33,41,64,65], CyA in three [41,56,65], MMF in four [33,41,56,65], SIR in one [64] and PRED in two [64,65].

All the studies that we reviewed showed that the treatment with cIMD was associated with changes in the intestinal microbiota with the only exception of one study performed in mice treated with SIR [58]. The effect on α-diversity was variable among studies with nine papers reporting no effect [42,44,47,52,53,58,62,69,70], six an increase [46,50,51,57,67,68] and two a decrease [45,61]; data on α-diversity were not available in six studies [34,43,48,49,63,66]. A decrease in the *Firmicutes*/*Bacteroidetes* ratio (either as directly reported or as evident from data on the relative abundance of these two phyla) was observed in three of the animal studies [46,48,62] (but in [48] TAC was given together with antibiotics), an increase in four studies [45,57,66,68], whereas no change was evident in six studies [42,44,53,61,67,70]; no data were reported in the remaining ten papers [43,47,49,50,51,52,58,63,67,70].

Figure 4 and Figure 5 show a synthesis of the findings of the papers listed in Table 2 on the changes in the microbiota, related to cIMDs administration, respectively, in animals and humans. In both figures, we reported the main changes at the genus level (or, when these data were missing, at the level of order or family) for each of the reviewed papers, and the genera involved were grouped according to their phyla. As evident from both these figures, not all the studies covered the whole repertoire of the main bacterial genera of gut microbiota. Therefore, in measuring the prevalence of the change in the abundance of a specific genus (or phylum) we only considered the number of papers reporting data on this specific genus or phylum. Using this approach and looking at the studies performed in animals (Figure 4), we observed that 7/9 (78%) of the reviewed studies reporting data on the respective phyla showed an increase in Proteobacteria, 25/42 (59.5%) an increase in *Firmicutes*, 10/19 (52.6%) a decrease in *Bacteroidetes*, 4/5 (80%) a decrease in *Verrucomicrobia*, and 2/4 (50%) a decrease in *Actinobacteria*. At the genus level, we could not identify any specific bacterial signature of cIMD-induced microbiota remodeling even though some changes were observed more frequently than others. Specifically, *Lactobacillus* was increased in 3 of 4 papers, and *Bacteroides* in 4 of 5, whereas *Clostridium* was decreased in 5 of 5, and *Akkermansia* in 3 of 4.

In the human studies, we observed, at phylum level, an increase in *Proteobacteria* in 5/10 studies (50%), and in *Firmicutes* in 19/23 (73%) studies, a decrease in *Bacteroidetes* in 6/7 (85%) studies, in *Verrucomicrobia* in 2/2 (100%) studies, and in *Actinobacteria* in 5/7 (71%) studies (Figure 5). As for animal studies, also the papers performed in humans did not disclose at the genus level an unique bacterial signature of cIMD effect on the microbiota. Nonetheless, some genera were affected more often than others. Specifically, *Streptococcus* was increased in 4 of 4 papers whereas *Blautia* and *Bifidobacterium* were decreased in 3 of 4, and *Akkermansia* in 2 of 2 (Figure 5).

Several factors could account for the heterogeneity in these studies of the findings on cIMD effects on microbiota and for the lack of a bacterial signature of cIMD effects on gut microbiota. They include the different species used and/or the different spontaneous or experimental disease statuses investigated. It is also to be considered that, as mentioned before, not all the papers reviewed report comprehensive data on gut microbiota composition and that, in some cases, the only information directly available is on few bacterial species on which the authors put their emphasis because of their potentially important pathophysiological role. An additional factor that makes difficult to produce an effective synthesis of the reviewed papers is that these studies used different cIMDs, sometimes in different combinations. To pinpoint drug-dependent variability we performed a separated analysis of the changes in the gut microbiota occurring when the various cIMDs were given alone and not in combination. Figure 6 shows the results of this synthesis for GCs, MMF, SIR and TAC, whereas CyA and EVERO could not be analyzed since they were studied only in one single paper. Due to the limited number of papers on each of the single cIMDS, we grouped the studies performed in humans and animals. When evaluating the results at the phylum level, the changes induced by the different cIMDs were similar. In fact, an increase in *Firmicutes* was observed in 66% of the studies with TAC that reported changes in this phylum, in 73% with GCs, in 75% with SIR and in 58% with MMF. Likewise, 100% of the studies with TAC that showed changes in *Proteobacteria* observed an increase in this phylum, whereas the percentage was 75% for GCs, SIR and MMF. By contrast, a decrease in *Bacteroidetes* was described in 67%, 86%, 50% and 60% of the studies, respectively, with TAC, GCs, SIR and MMF which reported changes in this phylum. When the data were, instead, at the level of genus no obvious drug-specific signature of drug-induced change in the microbiota could be identified since there was a large variability across the different studies. Nonetheless, some changes were recurrent in different papers. Specifically, in the case of TAC the two papers reporting data on *Allobaculum* both showed an increase in its abundance. Likewise, *Bacteroides* were increased in both the papers that measured its abundance, whereas *Akkermansia* was decreased in both the papers examining this genus. The most recurrent finding with MMF was a decrease in the abundance of *Clostridium* (observed in three papers out of three), whereas, with GCs, an increased abundance of *Ruminococcus* (3 of 4 papers) and *Lactobacillus* (2 of 2 papers) was observed, whereas *Bacteroidales* decreased in all the three papers measuring their abundance.

## 4. Discussion

In the present manuscript we reported the results of our systematic review of the evidence supporting the hypothesis that the intestinal microbiota affects cIMD pharmacology, whereas cIMDs affect the composition of the intestinal microbiota. The main finding of our study was that most of the published papers on these two issues corroborate the two aforementioned hypotheses. While this is true for the general picture of the reciprocal interrelationship between cIMDs and intestinal microbiota, the evidence becomes weaker for the subtopics that it covers.

Concerning the first point, 17/20 papers showed some effect of the intestinal microbiota on cIMD pharmacology, but only two papers were pharmacokinetics studies (moreover on two different drugs), only five were on toxicity (four of which on diarrhea and one on GvHD) and only six on cIMD efficacy (of which three on TAC and three on GCs). This fragmentation of the available evidence implies that the conclusions that can be drawn have to be considered preliminary and need further data to be confirmed. Moreover, evidence is missing for some cIMDs, i.e., SIR and EVERO. The hypothesis that cIMD pharmacology is affected by gut microorganisms should be placed in a more general context that assumes that the composition of the intestinal microbiota represents one of the factors responsible for the interindividual variability in patient response to drug therapy [108,109], together with pharmacogenomics with which it possibly interacts in a clinically significant manner [110]. As a matter of fact, one of the papers included in our systematic review showed that 13% of the 438 drugs successfully tested with a new method for the high throughput screening of drug metabolism by intestinal bacteria from human stools, were metabolized by intestinal microorganisms [26]. These metabolized drugs belonged to different therapeutic categories, including antiepileptic (clonazepam), antihypertensive (nicardipine and spironolactone), antipsychotic (risperidone), anticancer (capecitabine) and antiviral drugs (famciclovir). Amongst the cIMDs, GCs and MMF were positive, whereas CyA and EVERO were apparently not metabolized by human microbiota. The ability of the gut microbiota to metabolize GCs has been observed also in others of the studies that we reviewed [30,31]. The microbial enzyme involved has been identified with 17,20 desmolase which catalyzes the conversion of cortisol and its derivatives into 21-deoxysteroids, for instance, by converting hydrocortisone and hydrocortisone acetate in 20β-dihydrocortisone [26,111,112,113,114]. The evidence linking intestinal bacteria and 17,20 desmolase to corticosteroid metabolism is limited to in vitro testing. Two old papers, which were not retrieved in our search, showed that the urinary concentrations of 17-ketosteroids sharply increased after the intrarectal infusion of hydrocortisone and that this effect was abrogated by the oral administration of neomycin, suggesting that intestinal bacteria were involved [115,116]. However, according to our search, no formal pharmacokinetic study has been performed so far to assess the impact of this bacterial enzyme on the exposure to GCs and, therefore, the relevance of this pharmacomicrobiomic factor on the efficacy or tolerability of these immunosuppressive drugs still remains to be proved. Nonetheless it is worth to mention that high levels of (endogenous) 21-deoxysteroids have been found in patients with Cushing’s disease [117] and hypertension [118] and in vitro studies showed that they act as weak agonist on glucocorticoid receptors (GR) [119] and may transactivate the mineralocorticoid receptor (MR) although to a lesser extent than aldosterone and fludrocortisone [120]. These data suggest that microbial degradation of corticosteroids in the gut could generate metabolites that enhance their cardiovascular toxicity. The involvement of gut microflora in the pharmacokinetics of mycophenolic acid and of MMF has long been hypothesized to explain the enterohepatic recirculation of MPA, which, after being glucuronated to MPAG in the liver and released in the bile, is deglucuronated in the gut to generate MPA which is reabsorbed in the portal circulation [121]. Two papers included in the present systematic review demonstrated that the intestinal metabolism of MPAG is mediated by GUS producing bacteria [33,34]. In addition, the in vitro screening study by Javadan et al. [26], that we mentioned before, showed that bacteria from human stools may convert MMF into MPA. Importantly, the impact of intestinal microbiota composition on MMF and MPA has been demonstrated also in a pharmacokinetic study in humans, included in our systematic review [32]. This paper by Saqr et al., clearly showed in vivo, in patients who underwent HSCT that MPA enterohepatic recirculation and, consequently, MPA exposure were affected by intestinal GUS-expressing bacteria. Importantly, Simpson et al. [33] showed that the most important predictive factor for MPAG reactivation in the gut was the presence of GUS producing bacteria. The effect of intestinal bacteria on MPA pharmacokinetics could be clinically relevant as suggested by a study published after we finished our literature search, which showed a substantial decrease in MPA exposure in a very small case series of patients with signs of immunosuppressive therapy failure after starting an antibiotic treatment [122]. A paper by Guo et al. (2019) [25] examined the ability of intestinal bacteria to metabolize TAC and showed that *Faecalibacterium prausnitzii* converts this cIMD into a C-9 keto-reduction product, the M1 metabolite, which is approximately 15-fold less potent than the parent drug. These experiments were performed in vitro by incubating TAC with bacterial cultures; no formal pharmacokinetic study was performed even though M1 was measured in stool samples from healthy individuals and kidney transplant recipients. Preliminary data confirming the presence of M1 in the blood of patients receiving TAC were published as a letter by Guo et al. (2020) [123] but no pharmacokinetic investigation was performed. As discussed below, Lee et al. (2019) [28] demonstrated a correlation between the abundance of *Faecalibacterium prausnitzii* in the fecal microbiota and the dose of TAC required to attain therapeutic plasma concentration in the blood of patients with kidney transplantation, but, once again, no formal pharmacokinetic study was performed and M1 concentrations were not assessed neither in plasma or in stools. Recently, after we completed our literature search, a paper was published by Degraeve et al. (2023) [124] assessing the impact of the gut microbiota on TAC pharmacokinetics in vivo in mice. Whilst this study confirms that the gut microbiota profoundly affects TAC pharmacokinetics, the mechanism highlighted is completely different and consists in the decrease by bacterial released mediators in the activity of ABCB1 pumps, which extrude TAC form enterocytes and prevent their systemic absorption. In fact, upon antibiotic treatment, TAC exposure increased in a manner totally reversible with the ABC1B1 blocker zosuquidar. While the study by Guo et al. [25] seem to suggest that fecal microbiota reduces TAC bioavailability, the data reported by Degraeve et al. (2023) [124] seem to propend for the opposite scenario of intestinal bacteria enhancing TAC absorption. The inconsistent findings of these two studies (one performed in man and the other in mice) leave open the question of how the intestinal microbiota affects TAC pharmacokinetics and whether species-specific factors are involved. We retrieved only one paper by Zhou et al. that evaluated the effect of the intestinal microbiota on CyA pharmacokinetics [39]. The results of this investigation, which showed changes in CyA bioavailability depending on gut microbiota composition, contrast with the results of in vitro studies showing that this cIMD is not degraded in vitro by intestinal microorganisms [26,37]. A possible explanation for these apparently incongruent findings is that, in a similar way to what described by Degraeve et al. (2023) [124] for TAC, intestinal bacteria affect ABCB1. As a matter of fact, Zhou et al. [39] showed that intestinal microbiota disruption with antibiotic caused an increase in CyA bioavailability in rats that was accompanied by a decrease in protein levels of CYP3A1 and UGT1A1, which are involved in CyA metabolism in the liver and in the intestinal epithelium and ABCB1, the pump responsible for CyA efflux from hepatocytes and intestinal epithelial cells [39]. An additional factor (yet to be investigated) could be the ability (demonstrated in vitro in cultured intestinal cells [23]) of DCA and CDCA, generated by certain intestinal bacteria, to reduce the activity of ABCB1pumps, hence decreasing the efflux of CyA from the gut epithelium and, potentially, increasing its bioavailability.

Intestinal bacteria of the human microbiota may not only metabolize but also accumulate drugs such as duloxetine, montelukast, rosiglitazone and roflumilast that, after being internalized in their cytoplasm, become unavailable for systemic absorption with the final result of a reduced bioavailability and, possibly, clinical efficacy [16]. However, we did not find in our literature review any evidence that this also occurs with cIMDs.

Differences in the composition of intestinal microbiota were observed between GC-responsive and GC-unresponsive patients [40] and in patients requiring high or low TAC doses [28], suggesting that the efficacy of immunosuppressant therapy could depend on gut microbes. The most obvious explanation for such a correlation between intestinal microbiota and cIMD responsiveness could be that patients with different gut microorganisms had a different pre-systemic metabolism of these drugs. This hypothesis was not directly assessed for GCs in any of the papers that we reviewed. By contrast, a higher abundance of *Faecalibacterium prausnitzii*, the microorganism responsible for TAC metabolism, was observed in patients requiring TAC dose escalation [28].

Differences in the composition of gut microbiota were also observed between patients experiencing or not some cIMD toxicities, specifically diarrhea [24,29,33,38] and neutropenia [21]. The mechanism linking gut microbiota and diarrhea has been elucidated in the case of MMF and it is represented by the ability of certain bacterial species to produce GUS, deconjugate MPAG and, consequently, to increase the local concentrations of MPA which damages the intestinal mucosa [34,38]. Likewise, an increased abundance of the GUS-producing bacterium *Bacteroides dorei* has been observed, leading to a higher exposure to MPA in patients developing neutropenia when treated with MMF [21].

Concerning the second point under investigation, only one of the 32 papers investigating the effect of cIMDs on the intestinal microbiota had negative results. Therefore, as for the first point, i.e., the effect of the microbiota on cIMDs, the big picture of a bidirectional interrelationship between cIMDs and intestinal microbiota seems to be confirmed. At the phylum level, the most commonly observed finding, in both animal and in human studies, was a decrease in *Bacteroidetes* and *Verrucomicrobia*. The decrease in Firmicutes was more prevalent in human (85%) than in animal (53%) studies. At the genus level, however, we were unable to identify, an unique pattern of changes in the gut microbiota composition. Several factors may have contributed to the marked variability across studies including major differences in the species used, the disease conditions investigated and the administered cIMDs. The latter point would be of major relevance in the perspective of the present systematic review. However, based on our literature search, the evidence accumulated so far does not permit to identify drug-specific fingerprints of gut microbiota modifications. In fact, we identified only a limited number of papers investing the effects of single cIMDs. Most of them were animal studies, whereas, more cIMDs were generally used in humans, with the only exception of four studies in which a corticosteroid with no other drug was given to study participants without other cIMDs. Nonetheless, when we stratified the findings of reviewed studies without species constraints, we observed, for the different cIMDs, quite similar patterns at the phylum level, with an increase in the abundance of *Firmicutes* and *Proteobacteria* and a decrease in *Bacteroidetes*, *Proteobacteria* and *Verrucomicrobia*, but major differences in the genera involved. As described in Section 3 no microbiological signature of cIMD-induced alterations in gut microbiota emerged from our analysis even though some findings were in line with previous analyses, such as the increase in *Allobaculum* with TAC, the decrease in *Clostridium* with MMF or the increase in Lactobacillus with GCs [125]. 

Mechanistically, different factors could account for the ability of cIMDs to modify the composition of intestinal microbiota (see Gabarre et al., 2022 [125] for review). First, a wealth of experimental data show that non-antibiotic drugs may exert antibacterial effects in vitro and in vivo as clearly demonstrated, for instance for calcium channel blockers or antipsychotic drugs [126,127]. Such a mechanism could be relevant in the case of MPA, which, as a matter of fact, was originally identified as an antibiotic and not as an immunosuppressive drug [128,129], but does not seem to be relevant for the other cIMDs considered in the present systematic review. In fact, whereas a decrease in the severity of experimental mycobacterial infection was observed with the mTOR inhibitors, SIR and EVERO, this effect was due to the promotion of autophagy by this two cIMDs with no direct antibacterial toxicity [130,131]. Likewise, CyA and TACRO do not seem to have direct antibacterial effects even though they may exert an antifungal and antiviral activity, possibly by interfering with intracellular signaling cascades [132]. However, additional mechanisms may account for cIMD-induced changes in the intestinal microbiota, independently from a direct antibacterial effect. Specifically, the immunosuppressive activity of these drugs may interfere with the immune surveillance mechanisms that limit the proliferation of potentially pathogenetic bacteria whilst maintaining a condition of immune tolerance for commensal bacteria [133]. For instance, TACRO, GCs and the mTOR inhibitors decrease synthesis and release of the lectins RegIIIb and RegIIIg, two lectins with antibacterial properties [66]. The integrity of the mucosal barrier is another crucial factor in maintaining the normal composition of the intestinal microbiota, which may be altered by cIMDs. In fact, GCs decrease the synthesis and secretion of mucin and IgA [134,135], whereas MPA may directly damage gut epithelium loosening its tight junctions [136].

Another major point still to be clarified, since we did not find any paper directly addressing it, is how cIMD effects on the gut microbiota could impact on cIMD pharmacology. In other words, it remains to be established whether and to what extent the remodelling of gut microbiota occurring upon treatment with a given cIMD could promote the overgrowth (or, conversely, reduce the abundance) of bacterial species that metabolize or accumulate this specific cIMD. Such a phenomenon could be relevant in explaining changes in the efficacy of the immunosuppressant therapy occurring during treatment. In addition, considering that, as mentioned before, drugs belonging to multiple therapeutic classes besides immunosuppressants, the response to other, concomitant therapies could be affected as well.

The present systematic review has several important limitations. First, we considered both studies performed in humans and in rodents. Whilst this was largely a forced choice due to the limited number of papers published on cIMD pharmacomicrobiomics, it also implies potential problems in the interpretation of the results obtained. In fact, important differences exist in the composition of human and mice microbiota despite a 90% similarity at the phylum level and a 89% similarity at the genus level [137,138]. Another major limitation is the high heterogeneity of the experimental models that were used in the reviewed study. If we start considering the studies on cIMD pharmacology, a large fraction of them were just “pharmaceutical” investigations totally performed in vitro and exploring moiety degradation, only a few papers were performed in vivo and most of them were not specifically designed as “pharmacological studies” but, instead, they were clinical studies recording the occurrence of unwanted drug effect or therapy failure. Additionally, when we move to the studies aiming to investigate the effect of cIMDs on the intestinal microbiota, one of the major factor complicating and possibly biasing the interpretation of their results is the heterogeneity of the species and of the experimental models considered. As a matter of fact, only few studies, all performed in animals, tested the effect of selected cIMDs in healthy probands. On the contrary, in most of the animal studies and in all the studies performed in humans either diseases requiring cIMDS, such as organ transplantation or autoimmune diseases, or disease conditions induced by cIMDs such as obesity or diabetes were evaluated. Considering that the underlying disease may itself cause alterations in the intestinal microbiota as shown, for instance in chronic kidney disease (reviewed in [139,140] or in inflammatory bowel disease [141]), it is very hard to understand to what extent the modifications observed in its composition upon treatment with cIMDs are due to a direct effect of these drugs or to the improvement of the clinical condition requiring their use. Obviously, this conundrum is even harder to solve in the case of organ transplantation. For instance, in the case of liver transplantation, the important impact of the liver on the gut microbiota in the so-called gut–liver axis must be considered [142]. While animal studies adopted, in general, a controlled parallel group randomized design, most of the human studies were observational investigations and this raises further concerns about the reliability of the controls used in these studies.

## 5. Conclusions

The systematic literature review that we performed confirms and extends the findings of previous reviews and metanalyses showing that the gut microbiota is bidirectionally interrelated with cIMDs in a potentially clinically relevant manner [81,143,144,145]. The general conclusion emerging from all the evidence that we examined is that although pharmacomicrobiomics is a promising new tool to optimize the treatment with cIMDs, it is still in its infancy. Prospectively, one can imagine that, in the future, the dosage of cIMDs to be given to every single patient will be adjusted according to the composition of his/her intestinal microbiota and that further adjustment will be done during the treatment on the basis of changes occurring in gut microorganisms (even including those caused by the use of the same immunosuppressive drugs). However, a lot of experimental work remains to be done before this scenario can become clinical reality. In fact, most of the available evidence is limited to a small number of papers/patients and needs to be extended. In addition, data are still missing for some of the cIMDs considered such as SIR and EVERO. Even more importantly, the evidence that the composition microbiota has a causative role in determining the fate of transplanted organs or the prognosis of autoimmune disease is still limited and mostly related to animal models [146,147,148,149,150]. Likewise, it is still to be definitely demonstrated that interventions on the microbiota by the means either of probiotics supplementation or of genetic modification of resident gut bacteria may improve the prognosis in patients recipients of organ transplantation or affected with autoimmune diseases. With all these points still to be deeply explored we are hopefully at the beginning of a new exciting season of research advancement in the field of cIMD pharmacomicrobiomics. A final note concerning the data that have been reviewed in the present paper is that all the pharmacomicrobiomics interferences with cIMD pharmacology took place in the gut lumen and were strictly dependent on the oral administration of these drugs. This implies that, potentially, part of the pharmacomicrobiomic-related variability could be abated by using alternative administration routes and, in this perspective the use of new, emerging methods such as subcutaneous microneedles appears very intriguing [151,152]. 

## Figures and Tables

**Figure 1 biomedicines-11-02562-f001:**
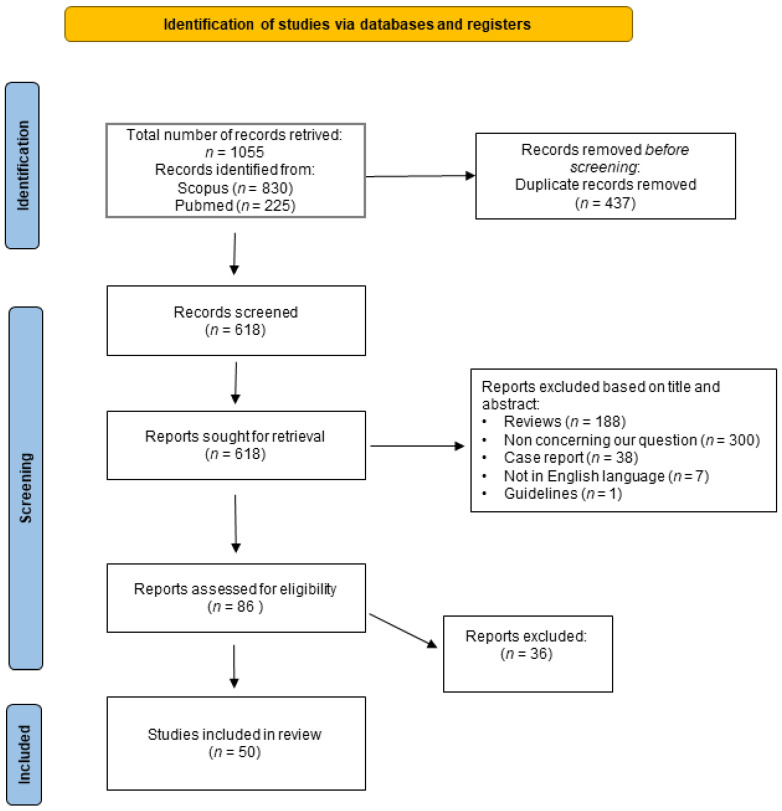
Flowchart of the systematic review. The figure shows the flowchart of this study prepared according to Page et al., 2021 [107]. For more information, visit: http://www.prisma-statement.org/ (accessed on 15 January 2023).

**Figure 2 biomedicines-11-02562-f002:**
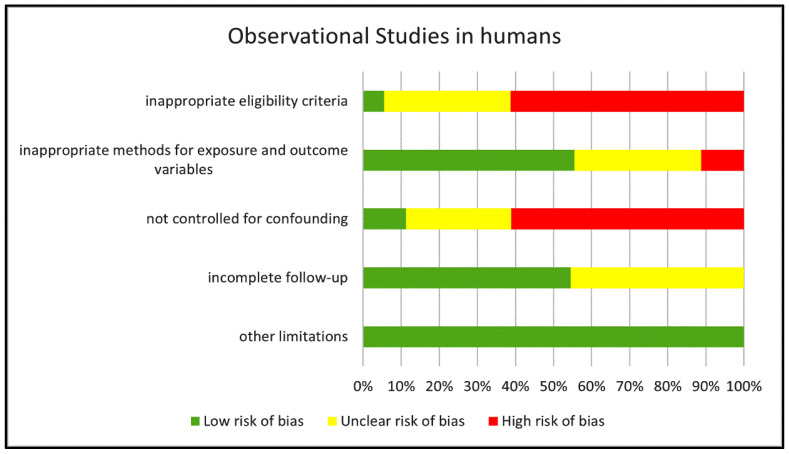
Risk of bias in the human observational studies examined. The chart reports the risk of bias in the reviewed human observational studies estimated for each of the GRADE categories and expressed as percentage of studies with “low”, “unclear” or “high” risk score.

**Figure 3 biomedicines-11-02562-f003:**
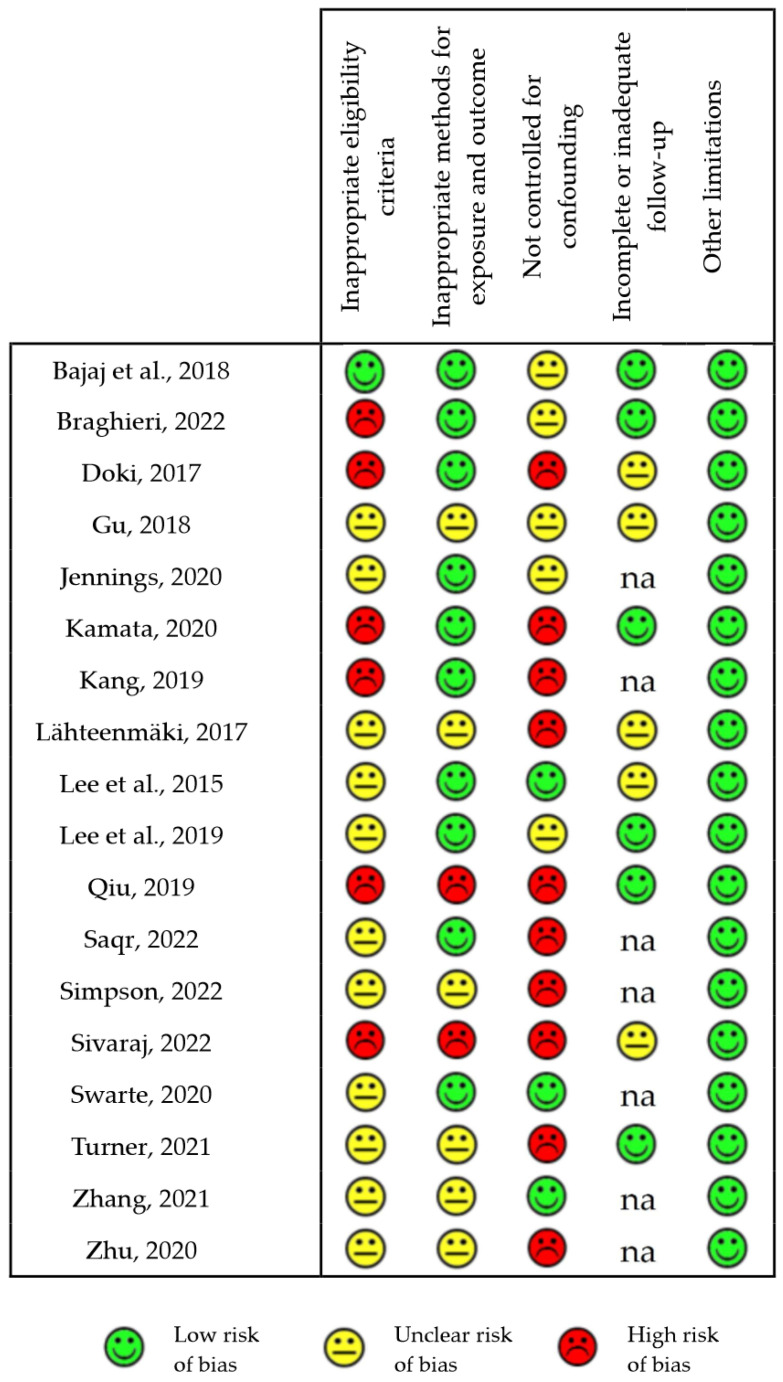
Risk of bias of observational human studies [21,22,24,27,28,29,32,33,35,38,40,41,54,55,56,60,64,65]. The chart reports the risk of bias for each of the individually estimated GRADE categories in all the reviewed human observational studies and classified as “low”, “unclear” or “high”. na: not applicable.

**Figure 4 biomedicines-11-02562-f004:**
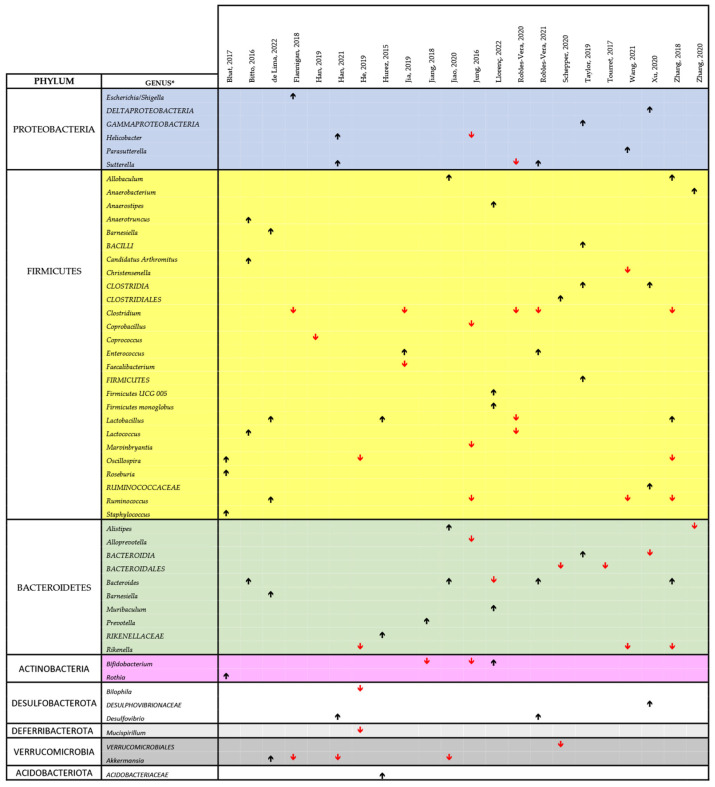
Comprehensive view of the effect of cIMDs on bacterial composition of the intestinal microbiota in animal studies [34,42,43,44,45,46,47,48,49,50,51,52,53,57,61,62,63,66,67,68,69,70]. The chart shows the main alterations in the composition of the intestinal microbiota at the genus level as identified in the text and/or tables of the respective papers. The black upward and the red downward arrows indicate respectively an increase or a decrease in the relative abundance of the respective bacterial genus. For reasons of space and to make the chart readable, we could not include all the data reported in each paper or in their supplementary information, but we have focused on those on which the authors of each publication emphasized. The study by Lyons et al., 2018 [59] was not included since no change in the composition of the intestinal microbiota was observed upon treatment of mice with SIR. * Whenever available in the original publications, the data reported are related to changes observed at the genus level. When this information was not available, we reported the data at the closest classification level (order or family).

**Figure 5 biomedicines-11-02562-f005:**
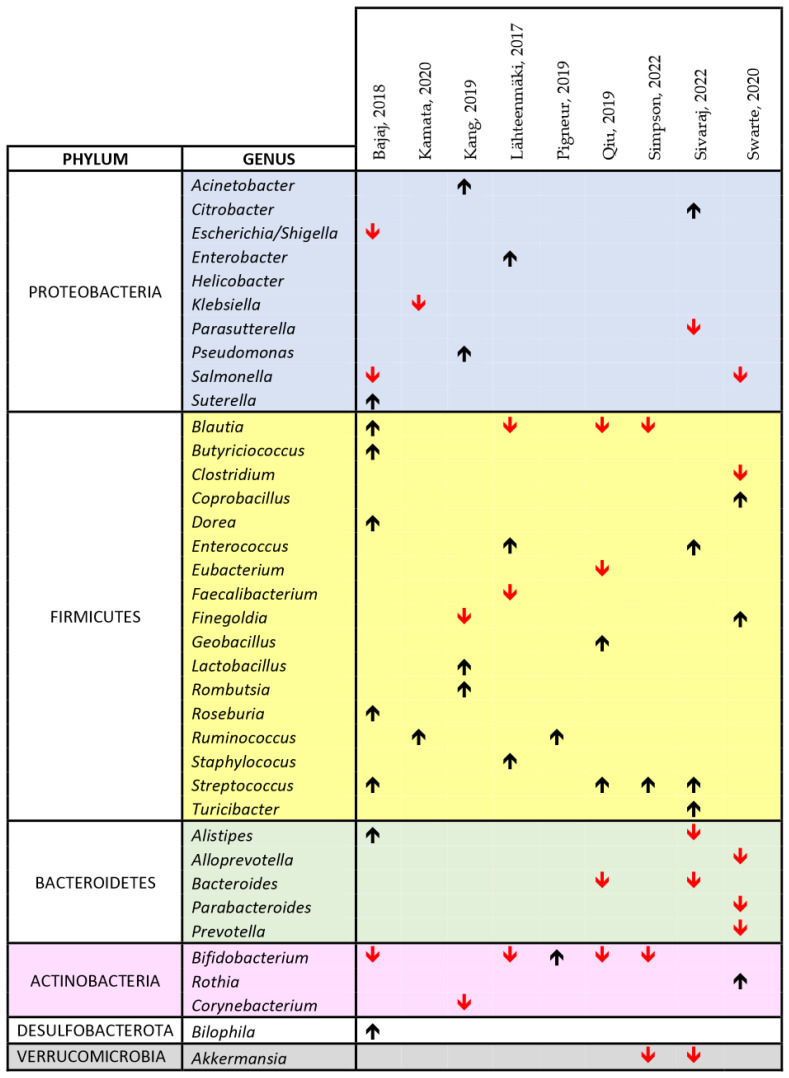
Comprehensive view of the effect of cIMDs on bacterial composition of the intestinal microbiota in human studies [33,41,54,55,56,59,60,64,65]. The chart shows the main alterations in the composition of the intestinal microbiota at the genus level as identified in the text and/or tables of the respective papers. The black upward and the red downward arrows indicate respectively an increase or a decrease in the relative abundance of the respective bacterial genus. For reasons of space and to make the chart readable, we could not include all the data reported in each paper or in their supplementary information, but we have focused on those on which the authors of each publication emphasized.

**Figure 6 biomedicines-11-02562-f006:**
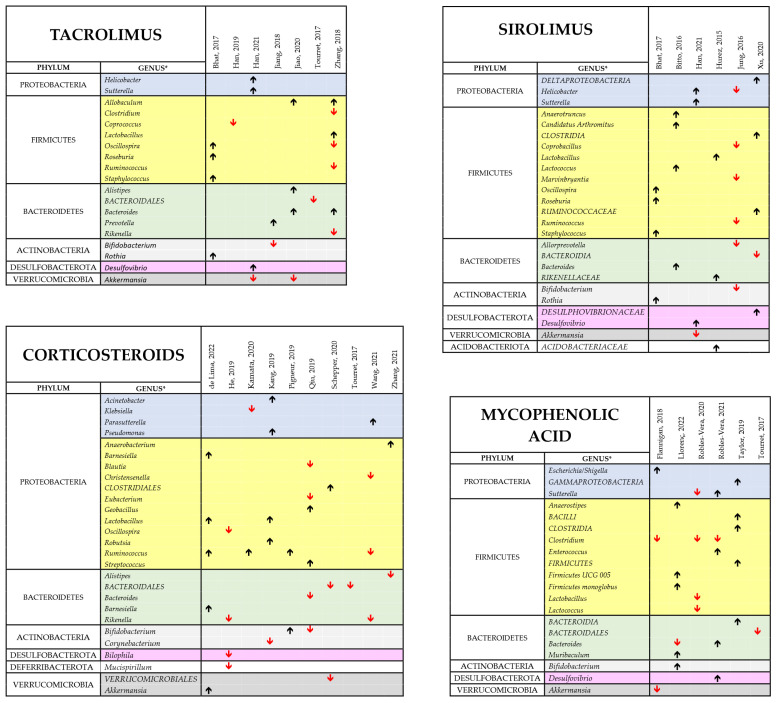
Comprehensive view of the effect of TAC, SIR, GCs and MMF bacterial composition of the intestinal microbiota [34,42,43,44,45,46,47,48,49,51,52,53,54,55,57,59,60,61,62,63,66,67,68,69,70]. The different panel show the main findings of the studies performed with each of the indicated cIMDs given alone to either experiment animals or human patients. The black upward and the red downward arrows indicate respectively an increase or a decrease in the relative abundance of the respective bacterial genus. No data have been reported for the other two cIMDs, CyA and EVERO since only one study was available for each of them. * Whenever available in the original publications, the data reported are related to changes observed at the genus level. When this information was not available, we reported the data at the closest classification level (order or family).

**Table 1 biomedicines-11-02562-t001:** Studies investigating the effect of intestinal microbiota on cIMD pharmacokinetics, efficacy and tolerability.

First Author/ Year/Ref.	Species	Tested cIMDs	Changes Observed (Yes/No)	Observed Effects
Braghieri et al., 2022 [21]	Humans, HT	MMF ± TAC	YES	No difference in α-diversity between neutropenic and non-neutropenic patients. Higher abundance in neutropenic patients of: *Acidaminococcus intestini*, *Agathobacter* spp., *Bifidobacterium dentium*, *Bacteroides dorei*, *Collinsella aerofaciens*, *Coprococcus comes*, *Lactobacillus fermentum*
Doki et al., 2017 [22]	Humans, aHSCT	CyA or TAC + MTX	YES	Higher *Firmicutes* abundance in the pre-transplant fecal microbiota of patients developing aGvHD
Enright et al., 2018 [23]	Humans, Cultured intestinal cell lines	CyA	YES	Increased CyA cytotoxicity by DCA and CDCA through the impairment of ABCB1 activity
Gu et al., 2018 [24]	Humans, KT	TAC + PRED ± MMF ± Mizoribine	NO	No change in CsA or TAC plasma concentrations after fecal transplantation in patients with diarrhea
Guo et al., 2019 [25]	Intestinal bacteria cultured in vitro Stools from KTR and normal donors	TAC	YES	TAC metabolism to the less active metabolites M1/M2 by *Faecalibacterium prausnitzii*. *Bacteroides cellulosilyticus*, *Bacteroides finegoldii*, *Bacteroides ovatus*, *Parabacteroides merdae*, *Parabacteroides johnsonii*, *Parabacteroides goldsteinii*, *Ruminococcaceae* sp., *Clostridium innocuum*, *Anaerostipes* sp., *Dorea formicigenerans*, *Clostridium clostridioforme*, *Clostridium hathewayi*, *Blautia* sp., *Clostridium aldenense*, *Clostridium symbiosum*, *Clostridium citroniae*, *Coprococcus* sp., *Clostridium bolteae*, *Clostridium cadaveris*, *Ruminococcus gnavus* and *Erysipelotrichales* TAC metabolism to M1/M2 upon culturing in vitro with fecal samples from KTR but not from controls
Javdan et al., 2020 [26]	Bacteria from human stool, cultured in vitro	BMET, CORTa, CyA, DEX, HCORT, HCORTa; MFA, MMF, mPREDN PRED, PREDN	YES	Drugs metabolized in vitro: BMET, CORTa, DEX, HCORT, HCORTa, mPREDN, PREDN, PRED, MMF Drugs not metabolized in vitro: CsA, MFA
Jennings et al., 2020 [27]	Humans, HT	Induction: mPREDN Maintenance: TAC + MMF + PRED	YES	Higher α-diversity in patients requiring high-TAC doses as compared with those on low-dose TAC. No difference in *Firmicutes*/*Bacteroidetes* ratio between high TAC and low TAC Higher abundance of 37 taxa in high TAC including *Anaesrostipes*, *Blautia*, *Lachnospiraceae* (uncharacterized), *Romboutsia*, *Roseburia*, *Subdoligranulum* and *Tyzzerella_4* Higher abundance one taxon of the genus *Bacteroides* in low TAC
Lee et al., 2015 [28]	Humans, KT	TAC + MMF ± PRED	YES	Higher abundance of *Faecalibacterium prausnitzii in patients* requiring TAC dose escalation Linear correlation between *Faecalibacterium prausnitzii* abundance and TAC dosing 1 week after transplantation
Lee et al., 2019 [29]	Humans, KT	TAC ± MMF ± PRED MMF ± TAC ± PRED	YES	Lower α-diversity in patients with post-transplant diarrhea Genus with higher abundance in patients with post-transplant diarrhea: *Eubacterium*, *Anaerostipes*, *Coprococcus*, *Romboutsia*, *Ruminococcus*, *Dorea*, *Faecalibacterium*, *Fusicatenibacter*, *Oscillibacter*, *Ruminiclostridium*, *Blautia*, *Bifidobacterium*, and *Bacteroides* Genus with higher abundance in patients with post-transplant diarrhea: *Escherichia*, *Enterococcus* and *Lachnoclostridium*
Ly et al., 2020 [30]	Bioinformatic analysis. Incubation in vitro of GCs with bacteria or recombinant enzymes	aTCORT, CORT, CORTN, DEX, FLUD, PRED, PREDN	YES	Identification by bioinformatic analysis of DesA/B in *Clostridium cadaveris*, *Intestinibacillus*, *Clostridium scindens*, and *Butyricicoccus desmolans* Degradation of aTCORT, CORT, CORTN, DEX, FLUD, PRED and PREDN by *Clostridium scindens* cultured in vitro and purified DesA and DesB
Qian et al., 2022 [31]	Enzymes cloned from bacterial strains	HCORT, TAC	YES	Identification in *Enterocloster bolteae* of three drug-degrading enzymes: DesE for nabumetone and hydrocortisone and TacA and TacB for TAC. Similar enzymes cloned from *Firmicutes*, *Proteobacteria* and *Bacteroidetes*
Saqr et al., 2022 [32]	Humans, HSCT	MMF + TAC	YES	No difference in α-diversity between high and low EHR (the ratio between MPA-AUC_4–8_ and MPA-AUC_0–8_) Higher abundance in patients with high EHR of *Bacteroides vulgatus*, *Bacteroides stercoris*, and *Bacteroides thetaiotaomicron* *Blautia hydrogenotrophica* abundance inversely correlated with EHR, the MPAG AUC_4–8_/AUC_0–8_ ratio, acylMPAG AUC_4–8_/AUC_0–8_ ratio, and MPA Css
Simpson et al., 2022 [33]	Humans, KT	TAC + MMF	YES	No difference in fecal microbiota between patients with or without MMF-induced diarrhea Higher rate of MMF reactivation from MPAG by fecal extracts of KTR as compared with control. Reactivation rate related to GUS and with the *Streptococcus parasanguinis* abundance
Taylor et al., 2019 [34]	Mice Humans, HT	Mice: MMF- ± Vancomycin Humans: MMF	YES	Mice: MMF increases GUS and MPAG conversion to MMF, Vancomycin reverses these effects Humans: stool GUS correlates with MMF exposure
Turner et al., 2021 [35]	Humans, aHSCT	PRED	YES	Higher abundance in patients with aGvHD who are PRED-non-responders, of *Bacteroides stercoris*, *Lactobacillus gasseri*, *Akkermansia muciniphila*, *Paraprevotella clara*, and *Sellimonas intestinalis* Higher abundance in patients with aGvHD who are PRED-responders of *Dorea longicatena* PRED refractoriness in patients with aGvHD predicted by the decrease in the *Dorea*/*Akkermansia* ratio
Vertzoni et al., 2018 [36]	Human fecal material, in vitro	Budesonide	NO	No significant budesonide degradation with either ileal or colonic simulated microbiota
Wang et al., 2015 [37]	Human fecal material, in vitro	CyA	NO	No significant CyA degradation in vitro
Zhang et al., 2021 [38]	Humans, KT	MMF ± PRED	YES	Lower gut microbial diversity and higher GUS in patients with diarrhea Lower abundance in patients with diarrhea of 12 bacterial genera: *Ruminococcus*, *Anaerostipes*, *Bifidobacterium*, *Eubacterium*, *Fusicatenibacter*, *Dorea*, *Ruminiclostridium*, *Oscillibacter*, *Gemmiger*, *Romboutsia*, *Streptococcus*, and *Coprobacillus*
Zhou et al., 2022 [39]	Rat	CyA ± ABX ± FT	YES	ABX increases CyA bioavailability and AUC; FT partially restores normal CyA bioavailability CyA AUC positively correlates with the abundance of *Akkermansia*, *Morganella*, *Parasutterella*, *Parabacteroides*, *Enterobacter*, *Escherichia*, *Shigella*, *Klebsiella*, and *Proteus* CyA AUC negatively correlates with the abundance of *Eubacterium Xylanophilum group*, *Desulfovibrio*, *Alloprevotella*, *Alistipes*, *Phascolarctobacterium*, *UCG 005*, *NK4A214 group*, and *Christensenellaceae R-7 group*
Zhu et al., 2020 [40]	Humans, UC	PRED	YES	In GC responders: higher abundance of *Enterobacteriaceae* and *Streptococcaceae* In GC-resistant patients: higher abundance at the family level of *Bacteroidaceae* and at the species level of *Clostridium perfringens*, *Corynebacterium durum*, *Roseburia unclassified* and *Ruminococcus sp_5_1_39BFAA* In GC-dependent patients: Higher abundance at the family level of *Clostridiaceae* and, at the species level of *Bacteroides vulgatus*, *Clostridium clostridioforme*, *Clostridium nexile*, *Coprococcus catus*, *Lachnospiraceae bacterium_1_4_56FAA*, *Parabacteroides merdae* and *Streptococcus gordonii*

Abbreviations: ABX: antibiotics; aGvHD: acute graft vs. host disease; aHSCT: allogenic hemopoietic stem cell transplantation; aTCORT: allotetrhydrocortisol; BMET: betamethasone; CORT: cortisol; CORTa: cortisone acetate; CORTN: cortisone; CyA: cyclosporin A; Des: 17,20-desmolase; DEX: dexamethasone; FT: fecal transplantation; GUS: β-Glucuronidase; HT: heart transplantation; FLUD: fludrocortisone; HCORT: hydrocortisone; HCORTa: hydrocortisone acetate; KT: kidney transplantation; KTR: kidney transplantation recipients; MMF: mycophenolate mofetil; mPREDN: methyl-prednisolone; MTX: methotrexate; PRED: prednisone; TAC: tacrolimus; UC: ulcerative colitis.

**Table 2 biomedicines-11-02562-t002:** Studies investigating the effect of cIMDs on the intestinal microbiota.

First Author/ Year/Ref.	Species	Tested cIMDs	Changes Observed (Yes/No)	Observed Effects
Bajaj et al., 2018 [41]	Humans, LT	Peri-operative: GC + MMF. Manteinance: TAC + MMF CyA + MMF	YES	Increase in α-diversity after LT Decreased abundance after LT of *Enterobacteriaceae* (*Escherichia*, *Shigella*, *Salmonella*) Increased abundance after LT of *Ruminococcaceae* and *Lachnospiraceae*
Bhat et al., 2017 [42]	Rats, normal	TAC or SIR	YES	No difference in α-diversity No difference in the *Firmicutes*/*Bacteroidetes* ratio Higher abundance in the SIR and TAC groups of *Akkermansia muciniphila*, *Roseburia*, *Oscillospira*, *Mollicutes*, *Rothia*, *Micrococcaceae*, *Actinomycetales* and *Staphylococcus*
Bitto et al., 2016 [43]	Mice, normal aging	SIR	YES	Higher abundance of segmented filamentous bacteria (*Candidatus Arthromitus* sp.) in the SIR group
de Lima et al., 2022 [44]	Rats, PTZ-kindling	PREDN	YES	No change in α-diversity No difference in the abundance of *Firmicutes* or *Bacteirodetes* Higher abundance in the PREDN 1 mg/kg group (but not in the 5 mg/kg group) of *Verrucomicrobia*, *Saccharibacteria* and *Actinobacteria* Higher abundance, at the family level, of *Porphyromonadaceae*, *Verrucomicrobiaceae* and *Clostridiaceae_1* in the PREDN 1 mg/kg and 5 mg/kg groups, and of *Erysipelotrichaceae* only in the PREDN 1 mg/kg group and of *Eubacteriaceae* in the PREDN 5 mg/group Higher abundance, at the genus level, of *Lactobacillus*, *Barnesiella*, and *Akkermansia* in PREDN 5 mg/kg and 1 mg/kg groups and of *Ruminococcus* only in the 1 mg/kg group Higher abundance, at the species level, of *Muribaculum intestinale* and *Akkermansia muciniphila* in the PREDN 5 mg/kg and 1 mg/kg groups and of *Saccharibacteria_genera_incertae_sedis TM7_phylum* only in the 5 mg/kg group
Flannigan et al., 2018 [45]	Mice, normal	MMF	YES	Lower α-diversity in the MMF group Lower abundance in the MMF group at the phylum level of *Bacteroidetes* and *Verrucomicrobia* and at genus level of *Akkermansia*, *Parabacteroides* and *Clostridium* Higher abundance in the MMF group, at the phylum level, of *Proteobacteria* and at the genus level of *Escherichia*/*Shigella*
Han et al., 2019 [46]	Mice, normal	TAC TAC ± ABX	YES	Increase in α-diversity in the TAC group partially reverted by ABX Decrease in α-diversity with TAC + ABX Lower abundance of *Verrucomicrobia* in the TAC group Higher abundance in the TAC + ABX group of *Verrucomicrobia*, family *Verrucomicrobiaceae*, genus *Akkermansia* Lower abundance in the TAC + ABX group of *Firmicutes*, family *Lachnospiraceae*, genus *Coprococcus* Lower *Firmicutes*/*Bacteroidetes* ratio in the TAC + ABX group
Han et al., 2021 [47]	Mice, normal	SIR	YES	Lower abundance in the SIR group at the phylum level of *Cyanobacteria*, *Firmicutes*, and *Verrucomicrobia*, at the family level of *Verrucomicrobiaceae* and, at the genus level, of *Akkermansia* Higher abundance in the SIR group at the phylum level of *Proteobacteria*, at the family level of *Helicobacteriaceae*, *Desufovibrionaceae* and *Alcaligenaceae*, and, at the genus level, of *Sutterella*, *Desulfovibrio* and *Helicobacter*
He et al., 2019 [48]	Mice, SLE (MRL/lpr mice)	PRED	YES	No difference in α-diversity. Lower abundance in the PRED group at the phylum level of *Proteobacteria* and *Deferribacteres*, and, at genus level, of *Rikenella*, *Mucispirillum*, *Oscillospira* and *Bilophila* Higher abundance in the PRED group at the genus level of *Prevotella* and *Anaerostipes*
Hurez et al., 2015 [49]	Mice, normal	SIR	YES	Minor differences in the composition of fecal microbiota Higher abundance in the SIR group of four taxa: *Lactobacillus Intestinalis* spp., and unclassified *Acidobacteriaceae* and *Rikenellaceae* (two taxa)
Jia et al., 2019 [50]	Rats, LT	CyA	YES	Higher α-diversity in the CyA group Higher abundance in the CyA group, in comparison with controls, of *Enterococcus* spp. Lower abundance in the CyA group, in comparison with controls, of *Faecalibacterium prausnitzii*, *Clostridium cluster XI*, and *Clostridium cluster XIV* Lower abundance in the CyA group, in comparison with the allograft group, of *Faecalibacterium prausnitzii*, *Enterobacteriaceae* spp., *Clostridium cluster I* and *Clostridium cluster XIV*
Jiang et al., 2018 [51]	Rat, LT	TAC	YES	Higher α-diversity in the TAC group Higher abundance of *Bacteroides-Prevotella*, *Enterococcus faecalis* and *Enterobacteriaceae* in the TAC group Lower abundance *Faecalibacterium prausnitzii* and *Bifidobacterium* spp. in the TAC group
Jiao et al. 2019 [52]	Mice, normal	TAC	YES	No difference in α-diversity Higher abundance in the TAC group of *Alistipes*, *Allobaculum*, and *Bacteroides* Lower abundance in the TAC group of *NK4A136*, *UCG-014*, and *Akkermansia*
Jung et al., 2016 [53]	Mice, DIO	SIR	YES	No difference in the Firmicutes/Bacteroidetes ratio Lower abundance in the non-obese SIR group as compared with non-obese control mice, at the genus level, of *Alloprevotella*, *Ruminococcus*, *Bifidobacterium*, *Marvinbryantia*, *Helicobacter*, and *Coprobacillus* Lower abundance in the obese SIR group as compared with non-obese control mice, at the genus level, of *Turicibacter*, unclassified *Marinilabiliaceae*, *Alloprevotella*, unclassified *Porphyromonadaceae*, *Ruminococcus*, *Bifidobacterium*, *Marvinbryantia*, *Helicobacter*, and *Coprobacillus*
Kamata et al., 2020 [54]	Humans, AIP	PREDN	YES	No PREDN-induced change in α-diversity PREDN-induced disappearance of *Enterobacteriales* (at the order level) and of *Klebsiellae* at the genus level PREDN-induced increase in the abundance of *Ruminococcus*
Kang et al., 2019 [55]	Humans, children with NS	PRED	YES	No change in α-diversity induced by PRED PRED-induced increase in the abundance of *Deinococcus Thermus* and *Acidobacteria* (at the phylum level), and at the genus level of *Romboutsia*, *Stomatobaculum*, *Cloacibacillus*, *Howardella*, *Mobilitalea*, *Deinococcus*, *Paracoccus*, *Stenotrophomonas*, *Gp1*, *Kocuria*, *Pseudomonas*, *Acinetobacter*, *Brevundimonas* and *Lactobacillus bacteria* PRED-induced decrease in the abundance of *Finegoldia* and *Corynebacterium*
Lähteenmäki et al., 2017 [56]	Humans, children with HSCT	CyA+ (MTX and MMF only in 1 patient)	YES	Higher abundance in HSCT, at phylum level, of *Proteobacteria*, and, at the genus level, of *Enterococcus*, *Staphylococcus*, *Enterobacter*, *Bacteroides* and unclassified genera of *Lachnospriracea* Lower abundance in HSCT, at phylum level, of *Firmicutes*, *Actinobacteria* and *Bacterodeites* and, at the genus level, of *Bifidobacterium*, *Bacteroides*, *Blautia* and *Faecalibacterium* (especially *F. prausnitzii*)
Llorenç et al., 2022 [57]	Mice, EAU	MMF	YES	Higher α-diversity in the MMF group Higher *Firmicutes*/*Bacteroidetes* ratio in the MMF group Higher abundance, at the genus level, of *Muribaculum*, *Bifidobacterium*, *Anaerostipes* and *Firmicutes UGC-005* in the MMF group compared with control mice Lower abundance, at the genus level, of *Bacteroides*, *Monoglobus*, *Eisenbergiella* and *Lachnospiraceae UCG-001* in the MMF group compared with control mice Higher abundance of *Lachnospiraceae NK4A136* in the EAU-MMF group compared with control EAU mice Lower abundance of *Lachnospiraceae UCG-001* in the EAU-MMF group compared with control EAU mice
Lyons et al., 2018 [58]	Mice, experimental colitis	SIR	NO	No change in fecal microbiota induced by SIR
Pigneur et al., 2019 [59]	Humans, children with CD	PRED	YES	PRED-induced a marginal increase in a-diversity PRED increased the abundance at genus level, of *Ruminococcus* and *Bifidobacterium*, and at species level, of *bacterium M62*, *A186*, *Faecalibacterium prausnitzii Roseburia intestinalis*, *Eubacterium* and *Bifidobacterium bifidum* PRED-decreased, at the genus level, the abundance of *Blautia*
Qiu et al., 2019 [60]	Humans, TM	PRED	YES	PRED decreased α-diversity At the phylum level, GC increased *Actinobacteria* and the *Firmicutes*/*Bacteroidetes* ratio At the genus level, GC decreased *Bacteroides*, *Bifidobacterium*, *Eubacterium* and increased *Streptococcus* and *Geobacillus*
Robles-Vera et al., 2020 [61]	Rat, DOCA salt hypertension	MMF	YES	Decrease in α-diversity (vs. DOCA-rats) Lower abundance in the MMF group, at the phylum level, of *Firmicutes* and, at genus level, of *Lactobacillus* and *Sutterella* Higher abundance in the MMF group, at the phylum level, of *Bacteroidetes*
Robles-Vera et al., 2021 [62]	Rats, SHR	MMF	YES	No effect on α-diversity The *Firmicutes*/*Bacteroidetes* ratio was higher in SHR than in control rats and it was normalized by MMF Higher abundance in the MMF group in comparison with SHR of the *Sutterella* genus Lower abundance in the MMF group in comparison with SHR of the *Clostridium genus* Higher abundance, at the phylum level, of *Firmicutes* and, at genus level, of *Lactobacillus* in the SHR group, normalized by MMF Higher abundance, at the phylum level, of *Actinobacteria* and *Bacteroidetes* in the SHR group, normalized by MMF
Schepper et al., 2020 [63]	Mice, GC-induced osteoporosis	PREDN	YES	Lower abundance of *Verrucomicobiales* and *Bacteriodales* in the PREDN group Higher abundance of *Clostridiales* in the PREDN group
Simpson et al., 2022 [33]	Humans, KT	TAC + MMF	YES	No difference in α-diversity between KTR and controls. Higher abundance in KTR, at class level, of *Gammaproteobacteria*, *Bacilli* and *Erysipelotrichia* Lower abundance in KTR, at class level, *Actinobacteria* and *Verrucomicrobiae*
Sivaraj et al., 2022 [64]	Humans, LT	TAC + SIR + PRED	YES	Higher *Firmicutes*/*Bacteroidetes* ratio in LTR Higher abundance in LTR at phylum level of *Firmicutes* and *Proteobacteria* and, at family level, of *Enterobacteriaceae*, *Erysipelotrichaceae*, *Fusobacteriaceae*, *Lactobacillaceae*, and *Veillonellaceae* at 3 months post-LT and of *Lachnospiraceae*, *Ruminococcaceae*, *Streptococcaceae*, and *Staphylococcaceae* after 6 months In comparison to pre-transplant samples, *Firmicutes* (in particular *Clostridiaceae*) were increased 3 months after LT and *Lachnospiraceae* and *Ruminococcaceae* 6 months post-LT
Swarte et al., 2020 [65]	Human, KT	CyA (18%) TAC (57%) AZT (9%) MMF (72%) PRED (96%)	YES	Lower α-diversity in KTR than in age-matched controls. Changes in α-diversity positively correlated with MMF use Higher abundance in KTR, at the phylum level, of *Proteobacteria*, and, at species level of *Escherichia coli* sp. Lower abundance in KTR, at the phylum level, of *Actinobacteria* and, at species level of *Bifidobacterium* sp., *Streptococcus termophylus*, *Blautia wexlerae* and *Streptococcus mitis* No difference in firmicutes
Taylor et al., 2019 [34]	Mice, normal	MMF	YES	Different fecal microbiota composition in the MMF group wih dominant bacteria represented, at the class level, by *Clostridia*, *Bacteroidia*, and *Bacilli* after 8 days of treatment and further expansion with continued MMF expansion of *Gammaproteobacteria*, *Erysipelotrichia*, and, to a lesser extent, *Deltaproteobacteria* classes Higher abundance in the MMF group of *Bacteroides vulgatus*, *Bacteroides fragilis*, *Bacteroides caccae*, *Bacteroides uniformis*, *Bacteroides ovatus*, and *Bacteroides nordii*
Tourret et al., 2017 [66]	Mice, normal	PRED TAC MMF EVERO PRED + TAC + MMF	YES	(1) FECAL MICROBIOTA: Higher β-diversity in the PRED group. Higher abundance, in the PRED group, at the phylum level, of *Firmicutes*. Lower abundance in the PRED group at the phylum level, of *Bacteroidetes*, and, at the order level, of *Bacteroidales*. Variable, unreproducble effects with other cIMDs (2) ILEAL MICROBIOTA: depletion of the *Clostridium* genus in the groups PRED andPRED + TAC + MMF
Wang et al., 2021 [67]	Mice, SLE (MRL/lpr mice)	PRED	YES	Higher α-diversity with PRED Higher abundance, in the PRED group, at phylum level, of *Proteobacteria*, and, at genus level, of *Parasutterella* and *Enterorhabdus* genus Lower abundance, in the PRED group, of *Rikenella*, *Christensenella*, *Ruminococcus*, and *Intestinimonas*
Xu et al., 2020 [68]	Mice, EAE	SIR	YES	(1) α-Diversity was decreased in EAE and normalized by SIR (2) Different composition of fecal microbiota with or without SIR (3) Most abundant bacteria in the control group: *Bacteroidia*, *Bacteroidetes*, *Burkholderiales*, *Sutterella*, *Anaerolinaceae.T78*, *Turicibacteraceae*, *Turicibacterales*, *Turicibacter* and *Bifidobacterium* (4) Most abundant bacteria in the EAE group: *Bacteroides*, *Bacteroidaceae*, *Rikenellaceae*, *Dorea*, *Mycoplasmataceae* and *Mycoplasmatales* (5) Most abundant bacterial spp in the SIR group: *Firmicutes*, *Oscillospira*, *Bacteroidales*, *Allobaculum*, *Anaerotruncus*, *Rikenellaceae.AF12*, *Odoribacteraceae*, *Odoribacter*, *Rikenella* and *Streptococcus*
Zhang et al., 2018 [69]	Mice, ST	TAC	YES	No difference in α-diversity Higher abundance in the TAC group of *Allobaculum*, *Bacteroides*, and *Lactobacillus* Lower abundance in the TAC group of Clostridium, *Ruminococcus*, *Rikenella*, *Ruminococcaceae*, and *Oscillospira*
Zhang et al., 2021 [70]	Rat, normal	PRED	YES	No difference in α-diversity Lower abundance in thePRED group, at the phylum level, of *Spirochaetes*, at the family level, of *Lachnospiraceae*, *Spirochaetaceae*, *Desulfovibrionaceae*, and *Rikenellaceae* and at genus level, of *Eisenbergiella*, *Alistipes*, and *Clostridium XIVb* Higher abundance in the PRED group, at the family level, of *Porphyromonadaceae*, and at the genus level of *Anaerobacterium*

Abbreviations: ABX: antibiotics; AIP: autoimmune pancreatitis; HSCT: hemopoietic stem cell transplantation; CD: Crohn’s disease; CyA: cyclosporin A; DIO: diet-induced obesity; EAE: experimental autoimmune encephalomyelitis; EAU: experimental autoimmune uveitis; EVERO: everolimus; KT: kidney transplantation; LT: liver transplantation; MMF, mycophenolate mofetil; PRED: prednisone; PREDN: prednisolone; SHR: spontaneous hypertensive rats; SIR: sirolimus; ST: skin transplantation; TAC: tacrolimus; TM: transverse myelitis.

## Data Availability

Not applicable.

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
