# Peer review of "Pharmacomicrobiomics of Classical Immunosuppressant Drugs: A Systematic Review"

_biomedicines, 2023, doi:10.3390/biomedicines11092562_

Round 1

Reviewer 1 Report

Overall, this is a well written systematic review publication and offers a good overview summarizing the state of knowledge concerning hypothetical role of gut microbiota affecting the pharmacology of classical immunosuppressant drugs. This manuscript shows rich content, providing a deep insight for some works: the study is within the journal’s scope, and I found it to be well-written, providing sufficient information. However, before publication some points need to be clarified.

My comments:

Line 132 – please remove outcomes from methodology. It should be moved to conclusions rather.

Line 314 – the authors should ensure that they used term “expression” in relation to genes only.

Line 309 – aHSTC acronym is used only one time in main text. I see no sense to abbreviate it.

Line 310 – the first appearance of “acute Graft vs Host Disease” is in line 309 and this is the place where it should be abbreviated.

Table 1, 2 and figures – bacteria’s names should be written in italics.

Line 360 – please explain what ABX stands for.

Author Response

REPLY TO REVIEWER 1

Line 132- Please remove outcomes from the methodology.

We included the outcomes in the methods section (paragraph 2.5) since it is one of the terms of the PICO (Population, Intervention, Comparison, Outcome) framework for evidence medicine and systematic reviews (Huang et al., AMIA Annu Symp Proc. 2006;2006:359-63. PMID: 17238363; PMCID: PMC1839740). To better clarify this point, we introduced a new sentence in the revised version of the manuscript at lines 111-112.

Line 314- The authors should ensure that they used term “expression” in relation to genes only.

As recommended by the reviewer the term expression has been replaced by more appropriate words, when not referred to genes. Specifically, at line 313 of page 8, we replaced “GUS expression” with ”GUS activity”, whereas, at line 560, page 22, we replaced “protein expression of CYP3A1” with “protein levels of CYP3A1”.

Line 314- aHSCT acronym is used only one time in main text. I see no sense to abbreviate it.

As requested we removed the aHSCT abbreviation from the main text.

Line 310- the first appearance of “acute Graft vs Host Disease” is in line 309 and this is the place where it should be abbreviated.

As requested we introduced the abbreviation for graft vs host disease (GvHD) where this term is first used.

Tables 1,2 and figures- bacteria’s name should be written in italics.

As requested the format of all the bacteria names has been changed to italics.

Reviewer 2 Report

1- Why is it important to investigate the role of gut microorganisms in the variability of clinical response to classical immunosuppressant drugs (cIMDs)?

2-Can you provide more details about the potential effects observed on cIMD pharmacokinetics, efficacy, or tolerability in the 17 out of 20 papers that evaluated this issue? Were these effects consistent or variable?

3-What specific findings were reported regarding the effects of tacrolimus, cyclosporine, mycophenolic acid, and corticosteroids on cIMD response?

4-Can you provide examples of specific intestinal bacteria whose relative abundance was significantly altered by cIMDs?

5-Were there any studies that explored the mechanistic basis for these microbiota changes?

6-Were there any limitations or potential biases in the reviewed papers that need to be considered in interpreting the results?

7- Based on the findings, what further research is needed to better understand this relationship and its clinical relevance?

8- The authors can use the following article to improve the discussion in the introduction: https://doi.org/10.3390/polym15092031

Author Response

REPLY TO REVIEWER 2

Why is it important to investigate the role of gut microorganisms in the variability of clinical response to classical immunosuppressant drugs (cIMDs)?

We thank the reviewer for raising the important point of the rationale behind investigating the role of microbiota in cIMD pharmacology. To address this question we added a new sentence in the Introduction (lines 95,96) and we modified the next sentence at lines 96-99. In addition, we return to this point  at lines 686-691, in the Conclusion section.  This new text highlights that the main reason of the interest of investigating the relationship between microbiota and cIMDs is that by manipulating the microbiota a better individualization of cIMD therapy could be achieved.

Can you provide more details about the potential effects observed on cIMD pharmacokinetics, efficacy, or tolerability in the 17 out of 20 papers that evaluated this issue? Were these effects consistent or variable?

To address this point we expanded the discussion of the manuscript in order to analyze more in detail the 17 papers mentioned by the reviewer and included in Table 1. In addition, we briefly examined a few papers related to the issue of microbiota effect on cIMD pharmacology that were not included in our systematic review either because non complying with our search conditions or because they were published after we closed our search. In detail, the new text is at lines 493-502, 518-526, 529-551 and 557-562. In support of the new text we added 5 new references (114, 115 and 121-123).

What specific findings were reported regarding the effects of tacrolimus, cyclosporine, mycophenolic acid, and corticosteroids on cIMD response?

Multiple cIMDs are combined to take advantage of their pharmacodynamic synergy (e.g. tacrolimus or CyA reducing IL-2 release and mTOR inhibitors impairing cellular response to this cytokine). This synergy increases the immunosuppressive potential of the treatment and, potentially, the effect on microbiota making virtually impossible to dissect the individual effects of the combined drugs.

Can you provide examples of specific intestinal bacteria whose relative abundance was significantly altered by cIMDs?

This has been done in the new version of the manuscript, both in the Results section (lines 388-392, 410-412 and 447-454) and in the Discussion (lines 611-615).

Were there any studies that explored the mechanistic basis for these microbiota changes?

A new paragraph was added to the discussion to address this important point raised by the reviewer (lines 616, 617 and 618-639) and 12 new references (125-136) were introduced in the new reference list.

Were there any limitations or potential biases in the reviewed papers that need to be considered in interpreting the results?

This important point raised by the reviewer was addressed by briefly discussing the limitations and risk of biases of each data synthesis in the Results section (lines 422-435) and by expanding the section of study limitation in the Discussion (lines 650-669). 

Based on the findings, what further research is needed to better understand this relationship and its clinical relevance?

This point raised by the reviewer, which we consider extremely relevant to give further perspective on the issue addressed in our systematic review, was addressed by adding new text in the Conclusion section of the manuscript (lines 684-691 and 694-708). Five new references were added as well in support of the new text.

The authors can use the following article to improve the discussion in the introduction:
https://doi.org/10.3390/polym15092031

Thanks for the suggestion. We agree that some of the problems arising from pharmacomicrobiomics factors are related to the oral intake of cIMDs and could be overcome by using alternative administration systems even including subcutaneous microneedles. This has been acknowledged at the end of the conclusion (lines 702-708) and by adding to the reference list the indicated paper and another one, more specifically related to CyA (refs. 152, 153).